# Raw-Cured Spanish Traditional Meat Product “*Chistorra de Navarra*”: Sensory and Composition Quality Standards

**DOI:** 10.3390/foods9081006

**Published:** 2020-07-27

**Authors:** María José Beriain, María Teresa Murillo-Arbizu, Kizkitza Insausti, María Victoria Sarriés, Inmaculada Gómez

**Affiliations:** 1Research Institute for Innovation & Sustainable Development in Food Chain, Campus de Arrosadía, Universidad Pública de Navarra, 31006 Pamplona, Spain; mariateresa.murillo@unavarra.es (M.T.M.-A.); kizkitza.insausti@unavarra.es (K.I.); vsarries@unavarra.es (M.V.S.); 2Departamento de Biotecnología y Ciencia de los Alimentos, Universidad de Burgos, 09001 Burgos, Spain; igbastida@ubu.es

**Keywords:** raw-cured meat product, sensory quality, chemical composition, fatty acid profile, *chistorra*

## Abstract

The aim of this work was to set the quality standards of the *chistorra de Navarra*, a raw-cured Spanish traditional meat product, through the study of its sensory and physicochemical features. The quality of *chistorra* samples, coming from 50 different artisan producers, were assessed during three sessions by expert assessors (*n* = 15). In the first session, instrumental colour (L*, a*, and b*) and appearance and odour parameters were evaluated in the raw products. In the second session, texture and flavour attributes were determined in cooked products. Finally, in the third session, the best 10 classified *chistorras* from the first and second sessions were sensorially evaluated and sampled for further analysis: texture (Warner Bratzler and texture profile analysis (TPA)), chemical composition, and fatty acid profile. The *chistorra*s with the highest sensory scores had high shear force values, flavour intensity, and fat/hydroxyproline ratio. The average fatty acid profile obtained for *chistorra de Navarra* was: 42% saturated fatty acids (SFA), 45% monounsaturated fatty acids (MUFA), and 13% polyunsaturated fatty acids (PUFA), which was similar to the one found in other raw-cured sausages. Considering the sensory evaluation, *chistorra* was defined as a product with an intense orange colour, and with high resistance value in the initial bite. It was also characterised by a high juiciness and tenderness, aroma, and meat flavour. In mouth, the pork fat, one of the ingredients of *chistorra*, was balanced without any of the ingredients dominating. Chemically, the *chistorra* was characterised by a fat content close to 67% (dry matter), low hydroxyproline occurrence (≤0.6), and protein amount ranging 18–38%.

## 1. Introduction

The production of cured products involves, in a general way, three well-defined phases: the mixing of ingredients and stuffing, fermentation, and drying (ripening). Physical, microbiological, and biochemical changes [1], involving tissue enzymes as well as microbial enzymes, take place in sausages during fermentation and drying. These changes are influenced by the raw material characteristics [2] and the process conditions [3,4], and they will have a direct effect on the organoleptic properties of the final product.

The *chistorra*, a raw-cured sausage, is traditionally elaborated in Navarra (Spain) from pork meat. Navarra is located at the north of Spain where *chistorra* consumption is high. The definition of *chistorra* and the average ingredients are defined in the quality standard for meat derivatives published by Ministerio de Agricultura, Alimentación y Medio Ambiente in 2014 [5]. The minced pork meat (6 mm size) is mixed with paprika and garlic and it is later stuffed into a thin casing (natural lamb casing or artificial). The drying process for raw sausages is highly variable in time and it is carried out under controlled humidity (75 to 80%) and temperature conditions (between 12 and 15 °C) [6]. The drying phase can last from two days to even three weeks, causing differences in the characteristics of the final product. The traditional *chistorra* elaboration does not include the incorporation of starter cultures because the raw meat microbiota, and the one coming from the manipulation, contribute to the microbial ecology of the product. It is consumed fried, roasted, or cooked. Uniformity in stuffing, absence of stains and colour can be mentioned among the characteristics of the product that most influence consumer acceptability [7]. Its consumption extends to Spain and also abroad.

For the last 20 years, the Navarra Butcher’s Association has organized contests in order to promote *chistorra de Navarra*. In recent years, its increasing consumption has promoted the application for the quality brand Protected Geographical Indications (PGI) of the European Union to protect and differentiate this product from others, such as *longaniza*, whose characteristics are different and lead to confusion in the market. This PGI label will protect the name *chistorra de Navarra* as a specific product with unique characteristics, linked to its geographical origin, as well as to the traditional know-how. Currently, *the chistorra de Navarra* PGI labelling is in the approval process phase following the European Union indications.

In previous editions of the contest it was observed that the best scored *chistorras* showed certain heterogeneity in the final quality. Considering a wide set of variables, both sensory and instrumental, one of the aspects that aroused most interest was the texture, especially the degree of firmness, which revealed the existence of some heterogeneity in the main ingredients’ amounts that formed part of the formulation of each manufacturer. Regarding *chistorra*’s colour, the finalist *chistorras* had similar visual appearance although there were detected some differences in the instrumental luminosity and intensity redness. In recent years, following the market trend and consumer requirements, a decrease in fat proportion of this kind of products has also occurred, giving as a result a leaner, less juicy and less flavourful *chistorra*.

Few studies regarding *chistorra* characterization are available and those that already exist were published in Spanish journals and a long time ago [8,9]. On the contrary, publications on raw-cured meat products such as sausages and *chorizo*, which could be considered the most similar to *chistorra*, are numerous [10,11]. Concerning the future of this type of products, there are studies indicating the interest of developing healthier lipid profile of meat and derivative products [12,13] without an impact on their quality by adding other substances to the product or decreasing the total fat content. To achieve this, for example, the Council for Agricultural Science and Technology of USA (CAST) [14] and Cava et al. [15] modified the lipid composition of the diet of the animals used as source of raw material for back fat. Focusing on the formulation of the meat products, the lipid fraction proportion can be changed as this was done by Bloukas et al. [16] and Muguerza et al. [17] where animal fat was replaced by olive oil and/or pre-emulsified olive oil with soy protein isolate in Greek sausages and *chorizo Pamplona*, respectively. Severini et al. [18], Bloukas et al. [16] and Beriain et al. [19], developed technologically and sensorially acceptable *chorizo* using olive oil pre-emulsified with soy protein as substitute of pork fat, reaching a 33.3% replacement proportion in the studies of Severini et al. [18]. Similarly, this healthier lipid profile trend could be considered the future of the *chistorra*; however, the formulation changes may have a negative impact on quality and palatability and this will need further research. For this reason, rigorous studies on the composition and sensory quality of traditional products are necessary to maintain their traditional characteristics, original and genuine, inherited from manufacturing practices rooted in the history of the Navarrese people. Therefore, the aim of this work was to set the quality standards of the traditional *chistorra de Navarra*, a raw-cured Spanish traditional meat product, through the study of its sensory and physicochemical features.

## 2. Materials and Methods

Fifty different *chistorra* products presented at the regional contest were studied. They were fabricated by artisan producers of Navarra following the traditional formulation that determines the quality standard of this product [5]. That is, using pork meat and bacon as main ingredients, minced with salt, paprika and garlic, all kneaded, stuffed into a 30 mm diameter natural casing from lamb intestine and ripened for 3 days (25 °C, 90% relative humidity (RH)). Each *chistorra* was fabricated by a different producer and was representative of a big production batch of a minimum of 30 kg. The size of these meat companies was variable, from family businesses to companies with more industrialized technological processes, but in general, all of them can be considered as artisans.

### 2.1. Samples Selection Methodology and Design

The selection and evaluation method presented in this study, based on sensory attributes, has been carried out for twenty years at the regional contests. From the total of fifty different artisans, a 3 steps discrimination methodology, based on sensory quality, was carried out in order to choose the best 10 *chistorras* to whom the sensory and physicochemical characterization would be further performed, Samples were coded with a randomised 3-digit number and given to panellists one at a time in an order that was established to avoid the effect of sample order presentation, first-order or carry-over effects. For that, a jury of fifteen assessors performed the sensory evaluation of the *chistorra*. These assessors are part of the contest jury every year, and they have a valuable knowledge of the product [20]. The juries were distributed in 5 sections of 3 assessors each. The assessor profiles were butchers, academia and industry technicians, gourmets and gastronomy journalists. They are re-trained to evaluate the sensory attributes in 3 1-h sessions every year.

The stages of the experiment, with the selection of the samples and the analysis performed, are shown in Figure 1.

The first step consisted of the visual appearance evaluation of the 50 raw *chistorras* considering the following parameters: the presence of natural casing, tied craftsman, uniform filling, absence of stains, colour and texture (see Appendix A). Each panellist assigned a global value to the *chistorra* using an arbitrary scale rating from 0 to 20 points (being 20 the best grade). After that, the final global grade of each *chistorra* was obtained by averaging all the values assigned to each *chistorra*. Thus, the ten *chistorras* with the worst global score values, based on sensory parameters of the raw product, were withdrawn. On this first stage, the instrumental colour of the 50 raw-cured *chistorra* samples was also measured.

The second step consisted of the sensory evaluation of the cooked *chistorras* (40 samples selected at the first step). The two best-evaluated *chistorras* from each of the five panellist sections, were selected based on sensory parameters of the cooked products. Thus, at this second step, 10 *chistorras* were selected to pass on to the third and last step.

At the third step, the 10 best *chistorras* of step 2, were evaluated. A sensory evaluation of the cooked samples was performed. In addition to that, the raw *chistorras* were sampled for: composition, fatty acids analysis, microbiological analysis, Warner Bratzler Shear Force (WBSF) and texture profile analysis (TPA). Samples were kept at 5 °C until chemical composition and microbiological analysis were performed. Samples to be tested for instrumental texture and fatty acid profile were frozen at −20 °C until analysis, and they were thawed for 24 h at 5 °C before analysis.

### 2.2. Sensory Characterization.

In steps 2 and 3 the same sensory evaluation methodology was carried out on the cooked samples. Panellists ate unsalted toast bread and rinsed their palate with mineral water between samples. Each panellist rated samples for Texture, Flavour and odour, Residual flavour, Gristle absence, and Visual colour on a scale from 0 to 10 (being 10 the best grade). Therefore, the maximum score for each chistorra was 150 when adding the values of the three members of the session. At the third step of the selection process, the obtained score was added to the one obtained at the first step on the same raw chistorra. So, the final score of each chistorra included both sensory assessments (raw and cooked).

### 2.3. Raw Composition and Fatty Acids Analysis

For the *chistorra* chemical composition assessment, the following procedures were followed: ISO 1442-1973 [21] for the determination of moisture content; ISO 937-1978 [22] for the indirect determination of protein content; ISO 1443-1973 [23] for the determination of total fat content; Hill [24] for the determination of total hydroxyproline content and Boletin Oficial del Estado de España BOE-A-1979-21118 [25] for the determination of nitrate content. Briefly, samples were placed on a high temperature mechanical convection oven drying for moisture determination. Total protein composition was determined by the Kjeldahl method with a nitrogen conversion factor to protein of 6.25. The amount of fat was determined by fat extraction with a mixture of ethers after hydrolysis. Then, the solvent was removed by desiccation and a gravimetric determination was carried out. The determination of hydroxyproline was carried after protein hydrolysis in a strong acid medium and subsequent oxidation of this amino acid. This generates a hydroxyproline derivative which reacts with the p-dimethylaminobenzaldehyde reagent, giving a coloured compound that can be quantified at 560 nm using a spectrophotometer. The determination of nitrate consists in extracting a test portion, precipitating the protein and filtrating. The nitrates react with brucine providing a yellow-brown colour that can be photometrically measured at 410 nm. Three independent measurements were made on each sample. All these chemical composition measurements were determined by © Eurofins España (Pamplona, Spain). The analysis were performed in duplicate.

The determination of the fatty acid profile of the samples was performed, in triplicate per sample, using the fat extraction methylation procedure of Whittington et al. [26] with the modifications of Aldai et al. [27]. Before the separation and quantification of the methylated fatty acids, a 1/10 dilution in hexane was performed. Fatty acid profile analysis was carried out using an ionization gas chromatograph (Agilent Technologies GC 7890A system, Madrid, Spain) connected to an automatic sample injector (Agilent Technologies 7683B series, Madrid, Spain) and using a cross-linked polyethylene glycol capillary column (60 m × 0.25 mm internal diameter, 0.25 µm thickness, HP 19091N-136, Hewlett-Packard). The chromatographic conditions were as follows: initial column temperature 50 °C, programmed to increase at a rate of 3 °C/min up to 158 °C and then at 1 °C/min up to 165 °C, then at 2 °C/min up to 190 °C and then increasing again at 1 °C/min up to 198 °C, followed by a ramp of 0.25 °C/min until 205 °C, then 0.5 °C/min till 210 °C, then 1 °C/min to 222 °C, and finally 2 °C/min to a temperature of 240 °C. The injection and detector temperatures were maintained at 255 °C and 240 °C, respectively. Helium was used as carrier gas. Fatty acids were identified by comparison of their retention times with those of pure methylated standards and quantified using tricosanoic acid methyl ester (C23:0) as internal standard acquired to NuCheck Prep. Inc. (Elysian, MN, USA), and reported as percentage of the total fatty acids determined.

### 2.4. Microbiological Analysis

Microbiological analysis were carried out on the *chistorra* raw samples following the protocols from ISO 4833 [28], ISO R-7218 [29] and ISO 6887-2 [30]. For that, VIDAS Immuno Concentration Salmonella to test for salmonella, and VIDAS Listeria Monocytogenes to test for Listeria monocytogenes sp. were used following manufacturers’ instructions. The results were expressed as absence or occurrence. Three independent replications were made on each sample.

### 2.5. Instrumental Colour

The instrumental colour of the raw *chistorras* was measured using a portable Minolta CM-2002 spectrophotometer (Konica Minolta Business Technologies Inc., Tokyo, Japan). Five readings were obtained at non-overlapped points on the surface of each sample. Colour was measured in the CIELAB space [31], with standard illuminant D65, observer angle 10°. Colour values for lightness (L*), redness (a*), and yellowness (b*) values were measured. The spectrophotometer was standardized throughout the study using the standard white ceramic tile provided by the Konica company.

### 2.6. Instrumental Texture

Two instrumental texture analysis were performed using a TX-XT2i Micro Systems texturometer (Stable Micro Systems Ltd., Surrey, UK).

The Shear Force analysis was carried out using the WBSF probe according to the methodology proposed by Beltrán [32] and adapted to the *chistorra*. All samples were analysed in quintuplicate. Samples were tempered at room temperature and five cores of 3 cm length were cut per sample, and their inner diameter measured and recorded. The conditions of the analysis were crosshead speed 200 mm·min^−1^ and a 50 kg load cell, 40 mm distance, and calibration weight 10 kg. Full peak shear force was recorded and maximum shear force was calculated in kg as the mean of the five measurements.

The TPA was performed according to the methodology proposed by Mittal et al. [33]. All samples were analysed in quintuplicate. Samples were subjected to a two-cycle compression test with a cylindrical probe of 25 mm diameter and 50% compression ratio was used. Force-time deformation curves were derived with a 250 N load cell at a constant crosshead speed of 2.0 mm/s. The textural parameters were calculated: hardness (g), adhesiveness, cohesiveness (dimensionless), springiness (dimensionless), and chewiness (g).

### 2.7. Statistical Analysis

In the case of the physicochemical and microbiological assays, the means of the results from the replications were calculated. The statistical treatment of the data was subjected to one-way analysis of variance (ANOVA) and it was carried out with the package IBM SPSS Statistics version 24 (IBM Corp., New York, NY, USA). The descriptive parameters were calculated (mean, standard deviation, coefficient of variation, maximum and minimum) for each variable. Tukey’s test was performed in order to identify significant differences between groups (*p* < 0.05). In addition, the correlations between variables were obtained using the Pearson coefficient of correlation.

## 3. Results

### 3.1. Samples Selection and Sensory Analysis

Considering the sensory evaluation method explained in Section 2.1, the ten *chistorras* withdrawn at the first step showed lower appearance visual values than the forty *chistorras* that passed on to the second step. They also showed different instrumental colour values. In general, the 10 withdrawn *chistorras* were more orange, with a lack of uniformity in the stuffing and with the appearance of some darker areas in the dough that could be perceived through the natural casing (data not shown).

Table 1 shows the values of the raw and cooked sensory attributes of the 40 *chistorras* that went on to the second step of evaluation.

The results obtained for the samples in step 2 showed that the 10 best scored *chistorras* had a better texture, odour, flavour and colour (*p* < 0.001) and lower presence of gristle (*p* < 0.001), when compared to the group of non-finalist *chistorras*. Focusing on the ten finalist *chistorras*, the highest sensory score was obtained by producer B4 followed by B8 and B10 (Figure 2).

### 3.2. Chemical Composition and Fatty Acids Analysis

The ten tested *chistorras* complied with the fat and protein requirements for this type of products set in the Spanish State Official bulletin (BOE-A-2014-6435) [34]: fat ≤ 80% and protein ≥ 14% (Table 2). In general, the obtained results for the ten *chistorras* provided coefficients of variation below 25% except for nitrates quantification which demonstrated that the addition of this additive is highly variable among formulations. However, all the *chistorras* fitted the additives maximum residual dose allowed by the Spanish Ministry of Health (BOE-A-2002-3366) [35]. It should be noted that the percentage of fat of the ten *chistorras* presented a variability of less than 10%, which indicated the homogeneity in the use of this component in the formulations of the producers. This did not occur for the hydroxyproline percentage, which showed a variability above 20%.

The three-best scored *chistorras* (B4, B8, and B10) showed fat/hydroxyproline ratio values above 200 (237.50, 200.59, and 209.72, respectively), which was not obtained by the other samples. This is probably explained by the low hydroxyproline levels observed in these three samples (Table 2). The *chistorras* B4 and B10 showed chemical composition values in the range of the mean value (SD), whereas for sample B8 the moisture and protein values were low (lowest one for protein) and the fat value was the highest considering all the tested *chistorras*. This indicates that more fat than lean pork was used for the formulation of this *chistorra*. Nevertheless, B8 *chistorra* was globally scored with a 570 value (third best position) at the sensory analysis.

Table 3 shows that there is no relationship between the chemical composition and the sensory evaluation, except between the content of collagen (hydroxyproline content) and the gristle content score, so that those *chistorras* with high hydroxyproline content obtained a lower score for gristle content. It can be seen that the three finalist samples had a higher fat content and, in turn, a lower hydroxyproline content (close to half) than the average value of the group of *chistorras*. Those samples with less visual gristle content showed less hydroxyproline levels which is correlated with the collagen occurrence in the samples.

The fatty acid profile of the ten *chistorras* evaluated in step 3, showed variability in the saturated fatty acids (SFA), monounsaturated fatty acids (MUFA) and polyunsaturated fatty acids (PUFA) among the different commercial brands (Table 4). The predominant MUFA was oleic acid (C18:1c-9) (42.13%), being also the most prevalent individual fatty acid in the *chistorras*. PUFA content was 13.20%, being linoleic acid (C18:2*n*-6) (12.14%) the most abundant one.

### 3.3. Microbiological Analysis

Microbiological results obtained for the raw *chistorras* at step 3 were negative for the presence of *Salmonella*.

Considering Listeria occurrence, the results were positive for half of the tested samples. However, this bacterium is thermosensible and should disappear when chistorra is fried or roasted to be consumed [37]. Considering that nitrates are added to control microbiological growth on this types of products, the current trend of reducing the nitrates salts in the food production and the results obtained on this study, this additive addition should be controlled due to its relationship with microbiological contamination.

### 3.4. Instrumental Colour

Table 5 shows the values of the colour coordinates L*, a*, b* of the raw samples of the 40 *chistorras* that passed on to the second step and the 10 *chistorras* that were rejected at the first step.

The differences in each colour parameter were statistically significant (*p* < 0.05). This effect was observed for parameters L*, a* and b*, which means that the withdrawn *chistorras* (*n* = 10) were less luminous and had less red and yellow intensities.

Regarding step 2, the ten best scored *chistorras* provided instrumental colour values similar to those obtained for the other 30 *chistorras* that were withdrawn (Table 6). The only slight difference was obtained for a* parameter (red coordinate). In fact, the ten best considered *chistorras* showed lower red coordinate values than the 30 *chistorras* withdrawn in step 2.

When studying correlations between sensory values obtained in step 3 of the cooked *chistorras versus* the instrumental colour no association was founded for any parameter (data not shown).

### 3.5. Instrumental Texture

At step 3, once *chistorras* were tested for WBSF analysis, the highest shear force values were obtained for *chistorras* identified as B4 and B10 (Figure 3). These two *chistorras* were also the best scored by panellists at the sensory evaluation. It should by indicated that, even if B8 did not show a high WBSF value (Figure 3), it reached high sensory scores, ranking third position at the contest. This was probably due to the fact that this *chistorra* showed the highest fat composition from all the participants (Table 2) and it has been demonstrated that this ingredient has a deep impact on the flavour and odour and residual flavour attributes, that together with the absence of gristle (low hydroxyproline percentage, Table 2) contributes to get a high sensory score [37,38].

Results for the double compression texture analysis, or TPA, of the 10 *chistorras* that reached step 3 at the contest are reported in Table 7. Differences in the obtained results for hardness, adhesiveness, and chewiness were wide as it can be seen from the coefficient of variations.

When performing a correlation between these instrumental values and the sensory results coming from the panellist’s evaluation (Table 8), cohesiveness was the only parameter that showed a significant correlation with flavour and a trend to signification with sensory texture, residual flavour and cooked global score. Cohesiveness values for the ten best scored *chistorras* at step 3 are shown in Figure 4. It can be observed that all the *chistorra*s showed very close cohesiveness and that there was not a direct relationship between the 3 best selected *chistorra*s and the instrumental cohesiveness values.

## 4. Discussion

Texture and flavour are critical factors regarding the appraisal of quality of this type of products and this indicates that the fat quantity and quality affect the sensory characteristics of the final product [38,39]. The results obtained in the present work show that the following fabrication aspects affect the sensory quality of *chistorra*: components composition such as fat content and quality of the meat, and drying period, which affects fermentation. In this sense, a large variability in the parameters determining the sensory quality of the different artisan processors has been observed (individual data not shown).

Consumers associate the colour of products with their quality [40]; thus, colour will represent one of the key factors when choosing and accepting a *chistorra*. This fact, together with the development of abnormal colourations associated to the processing and preservation technologies, makes in evidence the need for objective colour measuring techniques. To do so, the measurement of colour coordinates by means of colourimeters or spectrophotometers is generally carried out. The *chistorras* withdrawn by their appearance at the first step (Table 5) showed a lower value of the coordinates L*, a* and b* compared to the *chistorras* that passed on to step 2. It means that the *chistorras* that were chosen were lighter (higher L*) and with a greater intensity of redness (a*) and yellowness (b*). The judges differentiated the *chistorras* by their raw appearance and these differences were in agreement with the instrumentally measured colour coordinates (Table 6). When fresh meat is cured by means of a mixture of sodium chloride and a small proportion of potassium nitrate, the red colour of the fresh meat stays during the curing process and the finished product has practically the same colour as the fresh meat [41,42]. Then, upon cooking, red colour becomes more intense. This phenomenon occurs in *chistorra*. The colour of cured meat and sausages depends essentially on the chemical modifications of the natural pigments of animal muscles with curing salts (nitrites and nitrates) [40]. This is a complex and slow process and the results can be visually and instrumentally discriminated, at least for *chistorra* product, as it has been demonstrated in this study.

At step 2, the expert panellists evaluated the different cooked products of the 40 artisan producers. The products in which their ingredients were separated more easily (fat and lean), were the ones scored with highest texture scores. As it has been observed, the instrumental cohesiveness is not only correlated to sensory texture, but with the rest of sensory parameters except from colour (Table 8). This characteristic cohesiveness occurs when the *chistorra* formulation includes an adequate amount of fat (66% referred to dry matter) and meat of high commercial quality from muscles with low collagen content, determined by a low content of gristle (collagen fibril bundles) (27%), and therefore of hydroxyproline (0.3%). The composition of *chistorras* B4 and B10 were very close in fat and protein percentages to the above indicated proportions (66.5% and 28.1% for B4, and 68.2% and 26.8% for B10, respectively).

Coefficients of variation for the fat content in *chistorras* (8.23) indicated a homogeneity in the amount of this component in all the producers. The mean fat content 30–35% based on the wet sample seems to be a guideline accepted by processors. Several authors have demonstrated that this guideline guarantees the acceptability and the final sensory quality of the products [39,43]. Previous studies revealed the negative effect that a decrease in fat content could have on the acceptability of this type of products, as excessively lean products were found to show less juiciness, lower flavour intensity and they were saltier on the palate.

Although the number of studies regarding the *chistorra de Navarra* is low, the present study demonstrates that the fat content and the lipid profile found for this product are within the range of previously reported values for raw-cured sausages [19,44]. These profiles match with the ones found by other authors in pork fat [15] and in traditional raw-cured sausages [17]. Furthermore, the fat content of the *chistorras* studied were within the range of the values observed by different authors [17,18]. It could be concluded that the different processors use variable total fat contents in their formulations and their fatty acid profile indicates some important differences in some of the fatty acids (Table 4). Pork fat is the most variable component in terms of composition and its fatty acid profile depends on different factors such as sex, composition of dietary fat, weight at slaughter, anatomical location, thickness of subcutaneous fat, etc. [45]. The observed variability might be due to differences in bacon composition, probably due to variations in the feeding diet of the animals.

Finally, the negative effect that residual nitrites cause on consumers’ health and the trend of the food sector towards a more natural diet free of artificial preservatives, have forced the meat industry to decrease the use of these additives. The study carried out by Jafari and Emam-Djomeh [46] concluded that the reduction in nitrite (from 120 mg/kg to 80 mg/kg) was possible if supported by the use of the concept “barrier technology”. Regarding the safety of nitrites and nitrates, both agencies, the European Food Safety Authority (EFSA) and the Scientific Committee of Spanish Agency for Food Safety and Nutrition, recommend a restriction of the use of these additives as much as possible, but without losing protection against *C. Botulinum*. It is indicated that with correct hygiene, the application of Hazard Analysis and Critical Control Points (HACCP) and the establishment of short storage times, meat products can be fabricated without nitrites if control measures are applied. In the present study, a low amount of nitrite was observed in the formulation of practically all the tested *chistorras* (Table 2).

## 5. Conclusions

This work has characterised and defined the sensory and compositional quality standards for *chistorra de Navarra*, a raw-cured Spanish traditional meat product. Fat percentage is important in the overall sensory acceptability of *chistorra*. Colour (intense orange), bite texture (high resistance) and mouth cohesiveness were found to be the 3 main quality criteria for *chistorra*. It was also characterised by its high juiciness and tenderness, aroma and meat flavour. In mouth, the pork fat was balanced without any of the ingredients dominating.

The characteristic physicochemical parameters of *chistorra* were fat content close to 67%, low hydroxyproline occurrence (≤0.6), and protein quantity ranging 18–38%. The nitrates addition was highly variable among artisans. Moisture in *chistorra* ranged from 34 to 55% depending on the drying time. The fatty acid profile of this raw-cured meat product is highly dependent on the pork meat and bacon qualities used for the fabrication. In this study, the obtained profile for *chistorra* was 42% SFA, 45% MUFA, and 13% PUFA. These results showed that, concerning chemical composition and fatty acid profile, the *chistorra de Navarra* was quite similar to other raw-cured products reported in the literature.

These results on physicochemical characterization, composition and sensory quality of the *chistorra de Navarra* provide a means to maintain its traditional and local characteristics that support the PGI denomination application.

## Figures and Tables

**Figure 1 foods-09-01006-f001:**
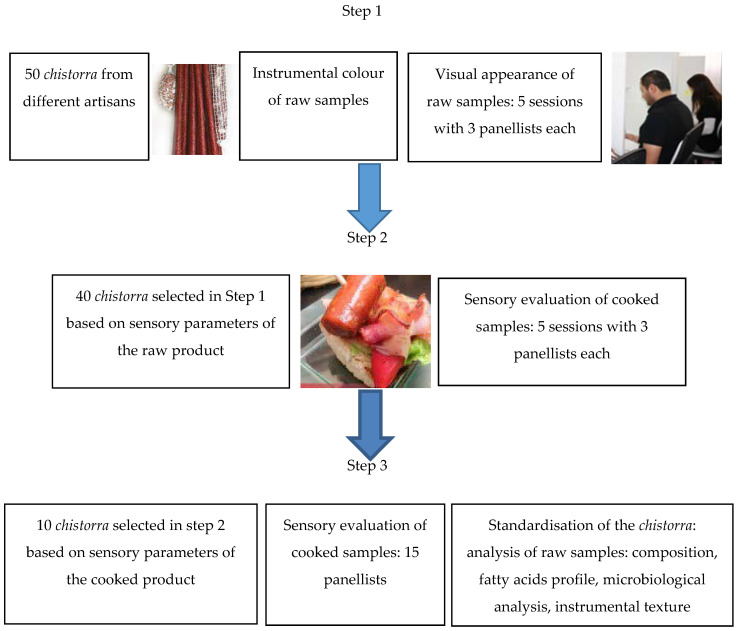
*chistorra* selection methodology.

**Figure 2 foods-09-01006-f002:**
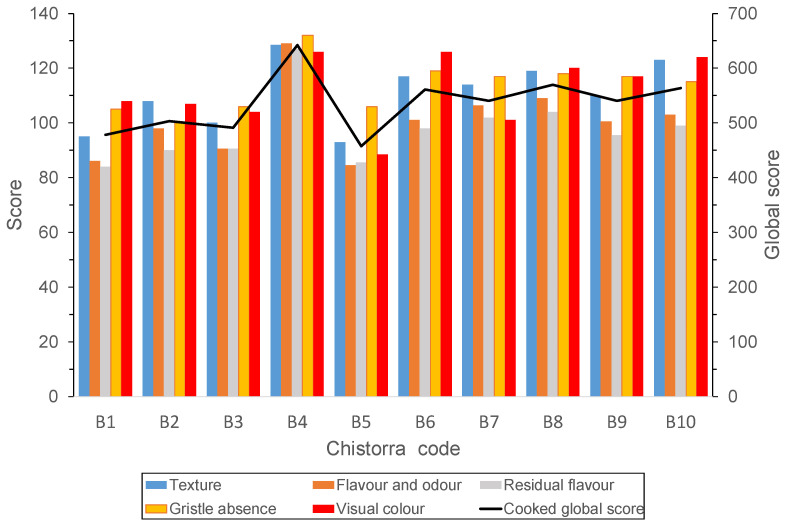
Results of the sensory evaluation for the *chistorras* in step 3 (*n* = 10).

**Figure 3 foods-09-01006-f003:**
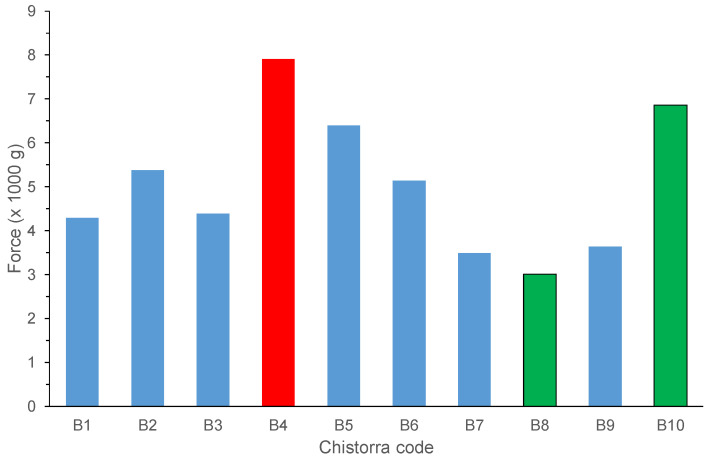
Warner Bratzler Shear Force texture results for the *chistorras* scored at step 3. Red bar refers to the winner *chistorra*; green bars, refer to the finalist ones.

**Figure 4 foods-09-01006-f004:**
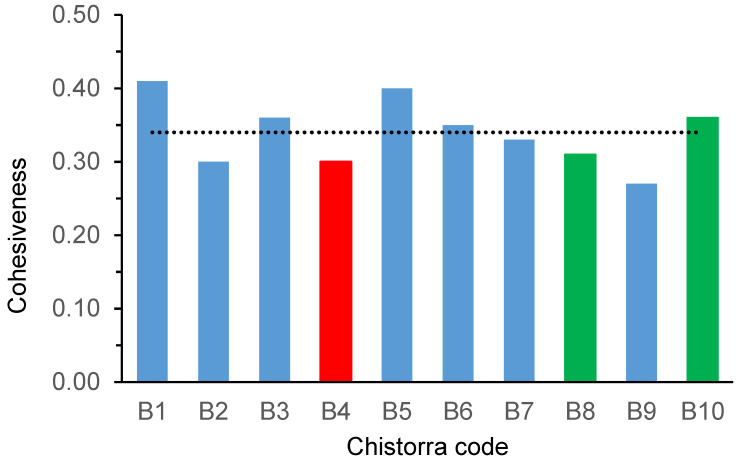
Cohesiveness measured by texture profile analysis (TPA) for selected *chistorras* at step 3. Red bar, refers to the winner *chistorra*; green bars, refer to the finalist ones; mean value is represented by a dotted line.

**Table 1 foods-09-01006-t001:** Means and standard deviation (SD) of sensory attribute values obtained for *chistorras* in steps 1 and 2 of the contest. Data were grouped in the 10 best scored and the 30 worst scored *chistorras*.

Step	Parameters	Lowest Scored *chistorra* (*n* = 30)		Highest Scored *chistorra* (*n* = 10)	
		Mean	Sd	Mean	Sd
1 (raw)	Raw global score	14.57	0.25	15.83	0.43
2 (cooked)	Texture	6.04	0.13	8.00	0.23
	Flavour and odour	5.72	0.14	7.46	0.25
	Residual flavour	5.54	0.14	7.20	0.24
	Gristle absence	6.53	0.14	8.00	0.25
	Visual colour	6.44	0.14	8.10	0.24
	Cooked global score	30.28	0.56	38.76	0.96

Note: in order to provide these results, the mean individual values from each assessor for each *chistorra* were positioned in ‘lowest scored’ or ‘highest scored’ groups and afterwards a mean value per group was obtained.

**Table 2 foods-09-01006-t002:** Raw composition of *chistorra* samples.

*chistorra* Code	Moisture (%)	Protein (%)	Fat (%)	Hydroxyproline (%)	Nitrates (mg/kg)	Far/Hydroxyproline Ratio
B1	50.2	37.6	56.3	0.30	20.3	187.7
B2	45.0	24.0	70.7	0.57	3.5	124.0
B3	33.9	19.6	73.4	0.55	25.0	133.5
**B4**	**45.4**	**28.1**	**66.5**	**0.28**	**14.3**	**237.5**
B5	55.4	33.3	61.0	0.42	38.9	145.2
B6	44.8	30.2	63.2	0.41	37.0	154.2
B7	47.4	27.4	67.1	0.37	9.9	181.4
**B8**	**36.0**	**18.2**	**75.5**	**0.36**	**37.2**	**209.7**
B9	44.2	24.2	70.7	0.42	111.2	168.3
**B10**	**46.9**	**26.8**	**68.2**	**0.34**	**18.9**	**200.6**
Mean	44.9	26.9	67.3	0.4	31.6	174.2
SD	5.9	5.6	5.5	0.09	28.9	34.1
CV	13.2	20.7	8.2	22.6	91.3	19.6
Range	33.9–55.4	18.2–37.6	56.3–75.5	0.28–0.57	3.5–111.2	124.0–237.5

SD: standard deviation; CV: coefficient of variation. B4, B8, and B10 are in bold because they were the 3-best scored *chistorras*.

**Table 3 foods-09-01006-t003:** Pearson’s correlation coefficients (*r*) between sensory evaluation scores and the raw composition analysis in the samples of the step 3 (*n* = 10).

Step	Parameters	Moisture (%)	Protein (%)	Fat (%)	Hydroxyproline (%)	Nitrates (mg/kg)
1 (raw product)	Raw “appearance”	−0.193	−0.102	0.084	−0.434	0.099
3 (cooked product)	Texture	−0.276	−0.355	0.394	−0.400	−0.097
	Flavour and odour	−0.244	−0.312	0.349	−0.439	−0.119
	Residual flavour	−0.215	−0.246	0.273	−0.495	−0.126
	Gristle absence	−0.123	−0.105	0.125	−0.643 *	0.152
	Colour	−0.361	−0.223	0.234	−0.397	0.126
	Cooked global score	−0.271	−0.277	0.306	−0.506	−0.023

* *p* < 0.05.

**Table 4 foods-09-01006-t004:** Mean, standard deviation (SD), coefficient of variation (CV) and maximum and minimum values of the fatty acid profile of *chistorras* in step 3.

Fatty Acid	Mean	SD	CV	Max	Min
C12:0	0.09	0.02	16.92	0.13	0.08
C14:0	1.43	0.09	6.43	1.65	1.34
C14:1*c*9	0.03	0.01	20.92	0.04	0.02
C15:0	0.06	0.01	20.07	0.08	0.04
C16:0	24.63	1.03	4.36	26.50	23.17
C16:1*t*9	2.31	0.32	13.98	2.66	1.80
C17:0	0.34	0.06	16.58	0.46	0.25
C17:1*c*9	0.31	0.05	15.13	0.36	0.21
C18:0	12.71	1.21	9.62	14.71	10.83
C18:1*c*9	42.13	1.75	4.21	45.55	39.54
C18:2*n*-6 (LA)	12.14	2.44	20.93	16.32	8.53
C18:3*n*-6	0.08	0.04	44.29	0.16	0.03
C18:3*n*-3 (ALA)	0.16	0.08	50.15	0.25	0.01
C20:0	0.66	0.15	23.20	0.81	0.40
C20:1*c*11	0.06	0.03	59.89	0.16	0.01
C21:0	0.74	0.12	14.04	0.84	0.51
C20:2*c*11,*c*14	0.48	0.10	20.62	0.64	0.36
C22:0	0.09	0.03	28.67	0.12	0.03
C20:4*n*-6	0.40	0.06	15.35	0.54	0.32
C23:0	0.96	0.19	18.59	1.34	0.67
C22:2*c*13,*c*16	0.10	0.02	16.47	0.14	0.08
C24:1	0.08	0.02	22.89	0.11	0.05
SFA	41.72	2.18	5.40	44.55	38.12
MUFA	44.92	2.03	4.58	48.75	41.87
PUFA	13.20	2.55	20.04	17.67	9.41
*n*-3	0.16	0.08	50.15	0.25	0.01
*n*-6	12.62	2.47	20.35	16.92	8.94
PUFA/SFA	0.32	0.07	24.04	0.45	0.21
AI	0.52	NA	NA	NA	NA
TI	1.32	NA	NA	NA	NA

SFA: Sum of saturated fatty acids: C12:0 + C14:0 + C15:0 + C16:0 + C17:0 + C18:0 + C20:0 + C21:0 + C22:0 + C23:0; MUFA: Sum of monounsaturated fatty acids: C14:1*c*9 + C16:1*c*9 + C18:1*c*9 + C20:1*c*11 + C24:1; PUFA: Sum of polyunsaturated fatty acids: C18:2*n*-6 +C18:3*n*-6 + C18:3*n*-3 + C20:2*c*11,*c*14 + C20:4*n*-6 + C22:2*c*13,*c*16; *n*-3: Sum of *n*-3 fatty acids: C18:3*n*-3; n6: Sum of *n*-6 fatty acids: C18:2*n*-6 +C18:3*n*-6 + C20:4*n*-6. AI. Atherogenicity index. TI: thrombogenicity index. NA: non-applicable. AI and TI calculated according to Ulbricht and Southgate [36]: atherogenicity index: AI = ((C12:0 + 4 × C14:0 + C16:0))/(ΣMUFA +ΣPUFA-*n*-6 +ΣPUFA-*n*-3); thrombogenicity index: TI = (C14:0 + C16:0 + C18:0)/((0.5 × ΣMUFA + 0.5 × *n*-6PUFA + 3 × *n*-3PUFA + (*n*-3PUFA/*n*-6PUFA)).

**Table 5 foods-09-01006-t005:** Instrumental colour results of *chistorras* in step 1. Data were grouped in: 40 best scored *chistorras* (selected to pass on to step 2), and 10 worst scored *chistorras* (withdrawn).

Colour Parameters	*chistorras* Passed on to Step 2 (*n* = 40)		*chistorras* Withdrawn at Step 1 (*n* = 10)		Signification
	Mean	SD	Mean	SD	
L*	46.50	0.27	44.50	0.51	***
a*	28.79	0.20	27.33	0.38	***
b*	30.47	0.31	28.98	0.60	+

*** *p* < 0.001; + *p* < 0.1.

**Table 6 foods-09-01006-t006:** Instrumental colour results of *chistorras* in phase 2. Data were grouped in: 10 best scored *chistorra* (selected to pass on to step 3) and 30 worst scored *chistorras* (withdrawn).

Colour Parameters	*chistorras* Withdrawn at Step 2 (*n* = 30)		The Best *chistorras* Scored in Step 2 (*n* = 10)		Signification
	Mean	SD	Mean	SD	
L*	46.50	0.30	46.46	0.52	ns
a*	29.05	0.23	28.00	0.41	*
b*	30.47	0.34	30.44	0.59	ns

* *p* < 0.05; ns *p* > 0.05.

**Table 7 foods-09-01006-t007:** Results of texture profile analysis (TPA) from the ten best scored *chistorras* (step 3).

*chistorra* Code	Hardness (g)	Adhesiveness	Cohesiveness	Springiness	Chewiness (g)
B1	942.90	−151.92	0.41	1.00	385.76
B2	739.30	−176.50	0.30	1.01	222.30
B3	1457.35	−154.80	0.36	1.01	531.45
B4	1074.08	−86.41	0.30	1.01	325.50
B5	1622.87	−46.15	0.40	1.00	645.09
B6	1770.21	−28.86	0.35	1.01	622.48
B7	880.29	−220.37	0.33	1.01	287.97
B8	1113.87	−62.45	0.31	1.01	349.41
B9	951.42	−113.41	0.27	1.01	259.60
B10	1582.65	−157.86	0.36	1.00	570.59
Mean	1213.49	−119.87	0.34	1.01	420.02
SD	362.16	62.52	0.05	0.00	157.79
CV	29.84	−52.15	13.37	0.48	37.57

**Table 8 foods-09-01006-t008:** Pearson’s correlation coefficients (*r*) between sensory evaluation scores and the instrumental textural analysis (TPA and Warner-Bratzler shear force test WBSF) in the samples selected for step 3 (*n* = 10).

Parameter	Hardness (g)	Adhesiveness	Cohesiveness	Springiness	Chewiness (g)	WBSF (g)
Texture	−0.013	0.085	−0.604 +	0.285	−0.212	0.275
Flavour	−0.226	0.078	−0.666 *	0.457	−0.400	0.295
Residual flavour	−0.107	0.159	−0.550 +	0.403	−0.270	0.372
Gristle absence	0.076	0.330	−0.458	0.254	−0.099	0.249
Colour	0.091	0.262	−0.446	0.056	−0.093	0.177
Cooked global score	−0.047	0.190	−0.596 +	0.319	−0.241	0.298

* *p* < 0.05; + *p* < 0.1.

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
