# Peer review of "Raw-Cured Spanish Traditional Meat Product “Chistorra de Navarra”: Sensory and Composition Quality Standards"

_foods, 2020, doi:10.3390/foods9081006_

Round 1
Reviewer 1 Report
- english should be revised
- professional panellists...how do you determine they are professionist? trained?
- section 2.4 which parameter enumerated/researched? improve this section
moreover
- materials and methods:how many batches from each producer?
- generally, revise materials and methods as in many parts it is difficult to read
- results:should be revised (construction of sentences, english, clearness of exposition)
- results: no standard deviation is included? single data? in this case it is difficult with a single product for each industry to make some statements
- discussion: english should be revised as well as construction of sentences
Author Response
Rev1
Comments and Suggestions for Authors
- english should be revised
English has been thoroughly revised and the text has been reorganised in order to clarify the content
- professional panellists...how do you determine they are professionist? trained?
The assessor profiles were butchers, academia and industry technicians, gourmets and journalists and they were re-trained before each contest. This information has been included in the paper lines 115-116.
- section 2.4 which parameter enumerated/researched? improve this section
The microbiological results were expressed as absence or occurrence. Included in the text (line 193)
- Materials and methods: how many batches from each producer?
One sample from each producer which is representative of a big production batch of a minimum of 30 kg. This information has been included in the paper lines 102-103.
- generally, revise materials and methods as in many parts it is difficult to read
The text has been reorganised in order to clarify the content
- results: should be revised (construction of sentences, English, clearness of exposition
English has been thoroughly revised and the text has been reorganised in order to clarify the content
- results: no standard deviation is included? single data? in this case it is difficult with a single product for each industry to make some statements
Standard deviation has been included
Discussion: English should be revised as well as construction of sentences
English has been thoroughly revised and the text has been reorganised in order to clarify the content.

Reviewer 2 Report
There are some typos throughout the document:
- Line 61-62 Need to delete space between last word and period :mentioned_."
- Line 146 need to separate "0to 10.."
- Line 158 Add and "s" after sample (plural)
Microbiological results:
- On lines 288-289 you stated that the presence of Salmonella was negative on samples that reached phase 3, however at the end you your discussion lines 431-433 you stated that one of the samples detected presence of salmonella. Are these results from samples different from phase 3?? You need to clarify this
- Paragraph (lines 290-293) needs to be rephrased. Listeria was positive on half of the samples and you are stating since Listeria is thermosensible, it "should dissapear" during further thermal treatments. This may be true but a reference should be mentioned here or on the discussion if you are stating this fact. The next sentence talks about the reduction of nitrates in food production which, again it is another hurdle to control microorganisms (you also explain this further in the discussion section), but sounds like you are explaining thermal reduction with addition of nitrates. Again combined hurdles are used in the food industry to control microorganisms but need to rephrase this paragraph to be understood better.
Author Response
Rev2
Comments and Suggestions for Authors
There are some typos throughout the document:
- Line 61-62 Need to delete space between last word and period: mentioned_."
Due to re-writing of some parts to polish English, this word is now in line 55 and corrected.
- Line 146 need to separate "0to 10.."
Authors have separated "0 to 10...” now in line 152 of the paper
- Line 158 Add and "s" after sample (plural)
Authors have added "s" after sample (plural) now in line 162 of the paper
Microbiological results:
- On lines 288-289 you stated that the presence of Salmonella was negative on samples that reached phase 3, however at the end you your discussion lines 431-433 you stated that one of the samples detected presence of salmonella. Are these results from samples different from phase 3?? You need to clarify this
Salmonella was negative on all samples that reached phase 3 and this is indicated in the paper in section 3.3, lines 295-296.
- Paragraph (lines 290-293) needs to be rephrased. Listeria was positive on half of the samples and you are stating since Listeria is thermosensible, it "should dissapear" during further thermal treatments. This may be true but a reference should be mentioned here or on the discussion if you are stating this fact. Reference included in line The next sentence talks about the reduction of nitrates in food production which, again it is another hurdle to control microorganisms (you also explain this further in the discussion section), but sounds like you are explaining thermal reduction with addition of nitrates. Again combined hurdles are used in the food industry to control microorganisms but need to rephrase this paragraph to be understood better.
English has been thoroughly revised and the text has been reorganised in order to clarify the content. This paragraph rephrasing now available in lines 299-302

Reviewer 3 Report
The manuscript is interesting and provides innovative information about “chistorra” a raw-cured Spanish traditional meat product from the Navarra region.
Be careful in typing: sometimes the space between 2 words is missing
I would reduce the length of the paper, sometimes is redundant (even in discussion).
It’s a pity that you did not analyzed also the worst “chistorra” to compare their chemical composition against the best one, in order to investigate which chemical factors influence the taste perception.
English should be improved to make the manuscript more fluid.
Line 47-71: please provide some references
Line 124: Replace him/her with “they” which is the gender-neutral English form.
353-363 reference is missing
374 “thus” with a capital letter
Author Response
Rev3
Comments and Suggestions for Authors
The manuscript is interesting and provides innovative information about “chistorra” a raw-cured Spanish traditional meat product from the Navarra region.
Be careful in typing: sometimes the space between 2 words is missing
It has been revised and amended
I would reduce the length of the paper, sometimes is redundant (even in discussion).
The text has been reorganised in order to reduce the length and clarify the content
It’s a pity that you did not analyzed also the worst “chistorra” to compare their chemical composition against the best one, in order to investigate which chemical factors influence the taste perception.
The authors agree with the referee.
English should be improved to make the manuscript more fluid.
English has been thoroughly revised and the text has been reorganised in order to clarify the content
Line 47-71: please provide some references???
References have been provided
Line 124: Replace him/her with “they” which is the gender-neutral English form.
Rewritten in lines 132-133.
353-363 reference is missing
References have been provided
374 “thus” with a capital letter
English has been thoroughly revised and the text has been reorganised in order to clarify the content. This word is not in the updated document.
